# Insights into substitution strategy towards thermodynamic and property regulation of chemically recyclable polymers

Yi-Min Tu[1], Fu-Long Gong[1], Yan-Chen Wu[1], Zhongzheng Cai [1]✉ & Jian-Bo Zhu [1]✉

The development of chemically recyclable polymers serves as an attractive approach to address the global plastic pollution crisis. Monomer design principle is the key to achieving chemical recycling to monomer. Herein, we provide a systematic investigation to evaluate a range of substitution effects and structure−property relationships in the ε-caprolactone (CL) system. Thermodynamic and recyclability studies reveal that the substituent size and position could regulate their ceiling temperatures ($T_c$). Impressively, **M4** equipped with a *tert*-butyl group displays a $T_c$ of 241 °C. A series of spirocyclic acetal-functionalized CLs prepared by a facile two-step reaction undergo efficient ring-opening polymerization and subsequent depolymerization. The resulting polymers demonstrate various thermal properties and a transformation of the mechanical performance from brittleness to ductility. Notably, the toughness and ductility of P(**M13**) is comparable to the commodity plastic isotactic polypropylene. This comprehensive study is aimed to provide a guideline to the future monomer design towards chemically recyclable polymers.

Progressive accumulation of polymer waste has caused serious environmental issues[1–6]. To address plastic pollution, chemical recycling to monomers (CRM) is believed to be a desired approach that allows polymers to be depolymerized into pristine monomers and repolymerized without a loss of material performance[7–23]. The thermodynamic manipulation of polymerization and depolymerization is a prerequisite to develop a chemically recyclable polymer system. The thermodynamic parameters of the enthalpy ($\Delta H_p^\circ$) and entropy ($\Delta S_p^\circ$) reflect the capability for modulating the thermodynamics of the polymerization and depolymerization process[7,22,24–28]. According to the Gibbs free energy equation ($\Delta G_p^\circ = \Delta H_p^\circ - T\Delta S_p^\circ$), the polymerization temperature is a straightforward factor for controlling the equilibrium direction. Generally, for a polymer system with $\Delta H_p^\circ < 0$ and entropy $\Delta S_p^\circ < 0$, a ceiling temperature ($T_c$) at standard state could be calculated at $\Delta G_p^\circ = 0$, where the polymerization process is favorable at temperatures below $T_c$ while depolymerization is favored above $T_c$.

This $T_c$ value represents the thermodynamic recyclability of the system[18]. Therefore, designing a monomer with a mild $T_c$ value for polymerization would provide a foundation for chemical recycling to monomers at practically operable conditions. The discovery that γ-butyrolactone (GBL) was capable of polymerization and subsequent depolymerization has motivated the development of new chemically recyclable polymer materials[29]. However, the synthesis of PGBL requires an undesirable low temperature (−40 °C), and PGBL exhibits limited thermostability because of its low $T_c$ feature ($T_c = −136$ °C at 1 M). Continuing monomer design based on GBL led to a paradigm shift in modulating polymerizability and tuning the polymer properties for this low $T_c$ system (Fig. 1)[28,30–34]. The introduction of *trans*-fused rings to GBL endowed the resulting systems with enhanced polymerizability ($T_c = 4$ °C for 4,5-T6GBL and 0 °C for 3,4-T6GBL at 1 M) and superior thermostability while maintaining complete recyclability[31,33,35]. The bridged-ring strategy (BiL, $T_c = 106$ °C at 1 M)

[1]National Engineering Laboratory of Eco-Friendly Polymeric Materials (Sichuan), College of Chemistry, Sichuan University, 29 Wangjiang Rd, Chengdu 610064, P. R. China. ✉e-mail: zzcai@scu.edu.cn; jbzhu@scu.edu.cn

enabled the compromise of the conflicting polymerizability, recyclability, and material properties. Furthermore, this strategy was proved to achieve the polymerization and orthogonal depolymerization of both the GBL with $T_c$ = 118 °C at 1 M and cyclohexene with $T_c$ = 66 °C at 1 M for BiL[30,32].

In contrast to the PGBL system with low $T_c$ values, a commercialized plastic poly($\varepsilon$-caprolactone) (PCL) produced by ring-opening polymerization (ROP) of $\varepsilon$-caprolactone (CL) was reported to have a $T_c$ > 2000 °C for 1 M[26,36,37]. Thermal degradation of PCL above 300 °C has been reported to yield the monomeric CL and its oligomers driven by the removal of CL in the system[38–44]. However, the ring-closing depolymerization of PCL to CL generally required rigid reaction conditions such as high temperature, high vacuo, or high catalyst loadings[45]. To achieve practically chemical recycling to CL, it's necessary to lower the $T_c$ values via monomer modification. In fact, many CL-based monomers have been reported for ROP toward functionalized polyesters[46–74]. These precedents revealed that ring size and degree of substitution of monomers could affect their polymerization thermodynamic equilibrium behavior. However, the relationship between $T_c$ and chemical recyclability was not specifically studied. The renaissance

of this fundamental study would provide monomer design principle for the development of chemically recyclable polymers.

Despite the recent advance in monomer design toward chemically recyclable polymers[8,24,27,29–34,75–87], a systematic investigation to evaluate a range of substitution effects and structure−property relationships are remained to explore. Consequently, this current work is aimed to provide a more detailed understanding of the factors that affect thermodynamics and material properties. Herein, we designed a series of monomers with a variety of substituent sizes and positions based on the structure of CL (Fig. 1). Notably, this series of CL-based monomers with increasing substituted size significantly reduced their $T_c$ values from 2060 to 241 °C and achieved complete recyclability. A facile spirocyclic acetalization approach allowed the designed monomers to polymerize under mild conditions and yield high-molecular-weight polymers ($M_n$ up to 494 kg/mol). These resulting PCL-based polymers demonstrated distinct chemical and physical properties, manifesting the potential of substituent regulation for property improvements. This detailed investigation of their structure−de/polymerizability and structure−property relationships could inspire future monomer design toward intrinsically chemically recyclable polymers.

## Results
### Substitution effect on polymerization thermodynamics
To elucidate the substitution effect on the thermodynamics of ROP of seven-membered lactones, a single-step Baeyer-Villiger oxidation of cyclic ketones was exploited to construct a library of substituted caprolactones (**M1**−**M7**) on large scales (>10 g) with high yields (69−96%). **M8** bearing a fused benzene ring was prepared by oxidative cyclization of 2,2'-(1,2-phenylene) diethanol in 82% yield on 8-gram scale. The polymerizability of **M1**−**M8** was next investigated using a 1−2 mol% zinc catalyst [(BDI)ZnN(SiMe₃)₂] (**Zn-1**)[88] in toluene-$d_8$ at an initial monomer concentration of 0.1−0.5 M (Fig. 2). Their thermodynamic data was acquired by monitoring the polymerization equilibrium changes over a temperature range of 40 to 70 °C by variable-temperature $^1$H NMR spectroscopy (Supplementary Figs. 108−115). Consequently, the standard-state thermodynamic parameters of $\Delta H_p°$ and $\Delta S_p°$ for ROP of **M1**−**M8** were summarized in Table 1 and their $T_c$ values at 1 M were also calculated. Monosubstituted **M1**−**M4** with the increase of steric hindrance from a methyl group to *tert*-butyl group displayed the $\Delta H_p°$ values of −16.2 ± 1.16, −14.0 ± 0.50, −14.8 ± 0.67, and −12.7 ± 0.75 kJ mol$^{-1}$, respectively (Table 1, entries 2−5). Contrastingly, their $\Delta S_p°$ values showed an increasing tendency from −14.9 ± 3.58 to −24.7 ± 2.00 J mol$^{-1}$ K$^{-1}$. As a result, a significant decrease in the calculated $T_c$ values for **M1**−**M4** at 1 M from 814 to 241 °C was observed in comparison with the previously reported $T_c$ value of 2060 °C for CL[36]. These data revealed that the alkyl substituent size has a powerful impact on polymerization thermodynamics. Beyond the substituent size, the substituent location appeared to have an important influence on polymerization thermodynamics as well. Compared with **M5** having geminal dimethyl groups, **M6** showed a clear increase in $\Delta S_p°$ value to −34.4 ± 4.57 J mol$^{-1}$ K$^{-1}$, which became a dominant factor to further reduce its $T_c$ value to 308 °C (Table 1, entries 6 vs. 7). Surprisingly, **M7**

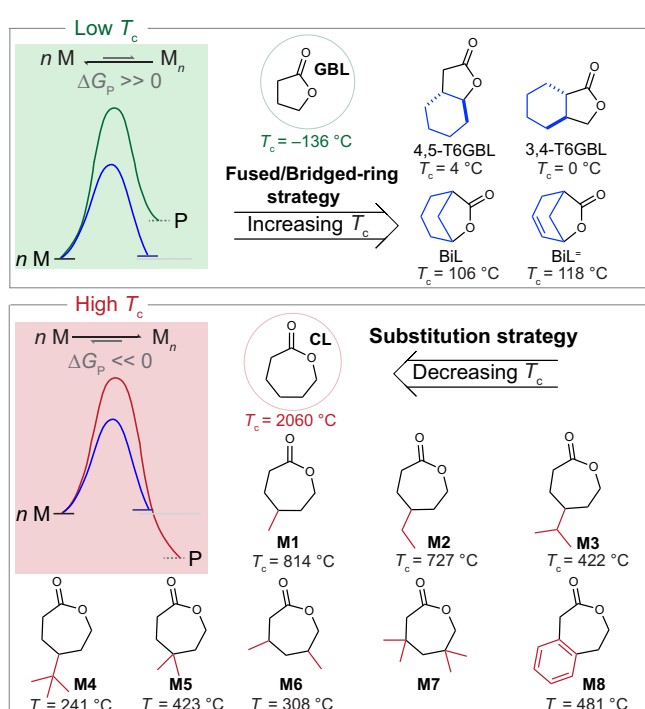

**Fig. 1 | Polymerization-depolymerization thermodynamic modification towards chemically recyclable polymers.** Fused/bridged-ring strategy has been applied to increase the ceiling temperature ($T_c$) for the PGBL system (relatively low $T_c$). In a PCL system with relatively high $T_c$, substitution strategy was investigated.

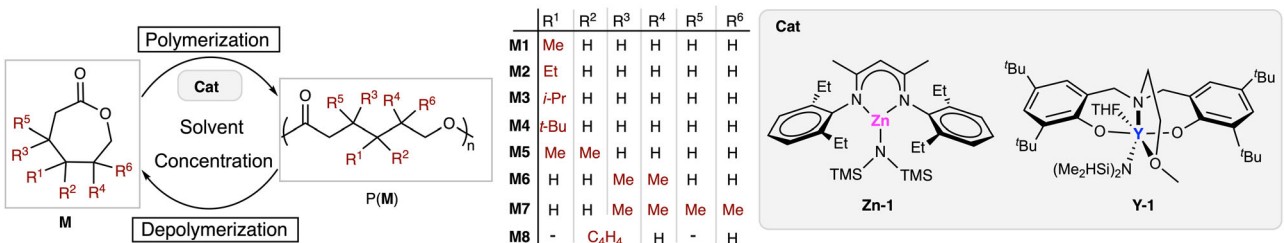

**Fig. 2 | Accessing chemically recyclable PCL-based polymers through ring-opening polymerization of substituted caprolactones.** Ring-opening polymerization of **M1**−**M8** and depolymerization of P(**M1**)−P(**M8**).

**Table 1 | Polymerization thermodynamic data for M1 – M8**

| Entry | M | $\Delta H_p°$ (kJ mol$^{-1}$) | $\Delta S_p°$ (J mol$^{-1}$ K$^{-1}$) | $T_c$ at 1 M (°C) |
|---|---|---|---|---|
| 1[a] | CL | −14.0 | −6.0 | 2060 |
| 2 | **M1** | −16.2 ± 1.16 | −14.9 ± 3.58 | 814 |
| 3 | **M2** | −14.0 ± 0.50 | −14.0 ± 1.50 | 727 |
| 4 | **M3** | −14.8 ± 0.67 | −21.3 ± 2.00 | 422 |
| 5 | **M4** | −12.7 ± 0.75 | −24.7 ± 2.00 | 241 |
| 6 | **M5** | −15.6 ± 0.42 | −22.4 ± 1.16 | 423 |
| 7 | **M6** | −20.0 ± 1.50 | −34.4 ± 4.57 | 308 |
| 8 | **M8** | −17.5 ± 0.42 | −23.2 ± 1.33 | 481 |

[a]Thermodynamic data were reported in ref. 36.

with geminal dimethyl substitutions on both β and δ positions disabled its polymerizability under the above condition. Attaching a non-adjacent aromatic ring to CL afforded **M8** with $\Delta H_p°$ of −17.5 ± 0.42 kJ/mol K$^{-1}$, $\Delta S_p°$ of −23.2 ± 1.33 J/mol K$^{-1}$, and $T_c$ of 481 °C at 1 M (Table 1, entry 8). Gratifyingly, **M8** performed improved air stability while maintaining a reasonable $T_c$ value in comparison to our previously reported nonadjacent aromatic monomer BDPO containing an extra heteroatom[75]. In comparison to substituent size, substituent numbers and positions that could contribute to more conformation and repulsion strain appeared to have a more pronounced influence on the monomeric ring strain and facilitate a large increase in the value of $\Delta H_p$. Substitution of CL could lead to a decrease in the flexibility of polymer chains, devoting significant loss in the conformational degrees of freedom for the resulting polymers. Consequently, the addition of substituents to CL was able to drive the thermodynamic equilibrium toward the depolymerization process and decrease the $T_c$ values of the resulting monomer systems. In the low-$T_c$ GBL system where both values of $\Delta H_p°$ and $\Delta S_p°$ were increasing, the change of $\Delta H_p°$ contributed to the final $T_c$ increase of the resulting polymer system. In contrast, the change in $\Delta S_p°$ appeared to be a predominant parameter in modulating the thermodynamics of polymerization and depolymerization for this high-$T_c$ CL system.

## Ring-opening polymerization studies

To execute the ring-opening polymerization, substituted caprolactones (**M1**–**M8**) were subjected to the catalyst **Zn-1** and an initiator p-tolylmethanol at the [monomer]:[catalyst]:[initiator] ([M]:[Zn-1]:[I]) ratio of 1000/1/1 in THF (Table 2). ROP of these monomers reached >70% conversions within 30 min except for **M7** (Table 2, entries 2–9). In comparison to the non-substituted CL (Table 2, entry 1), increasing the steric bulk of the substitution on the monomers diminished their polymerization activity. Beyond the substituent size, the substituent location appeared to have a negligible impact on polymerization activity. Compared with **M5** having geminal dimethyl groups, **M6** with two separated methyl groups on β and δ positions, exhibited a similar polymerization reactivity. Particularly, **M7** equipped with geminal dimethyl substitutions on both β and δ positions was unable to proceed with polymerization under a similar condition (Table 2, entry 8). The resulting polyesters P(**M**)s displayed the number-average molecular weights ($M_{n,SEC}$) values of 83.0–208 kg mol$^{-1}$ with narrow dispersity by size exclusion chromatography (SEC) analysis. These $M_{n,SEC}$ values of P(**M**)s were inconsistent with their corresponding theoretical values calculated from [M]$_0$/[I]$_0$ ratios and conversions ($M_{n,Calcd}$), demonstrating a *not* well-controlled polymerization system, since the high reactivity of these monomers led to undesired initiation events and inevitable chain transfer.

Driven by our thermodynamic and kinetic findings that prospective seven-membered lactones with the geminal disubstituent[75,80,89] could decreased $T_c$ values for ROP without sacrificing their polymerization reactivity, we expanded our geminal

**Table 2 | Ring-opening polymerization results of PCL-based monomers with Zn-1[a]**

| Run | M | Time (min) | Conv.[b] (%) | $M_{n,Calcd}$[c] (kDa) | $M_{n,SEC}$[d] (kDa) | Đ[d] ($M_w/M_n$) | $T_d$[e] (°C) |
|---|---|---|---|---|---|---|---|
| 1 | CL | 20 | 94 | 107 | 121 | 1.35 | - |
| 2 | **M1** | 30 | 72 | 92.3 | 93.7 | 1.18 | 366 |
| 3 | **M2** | 25 | 90 | 115 | 137 | 1.43 | 338 |
| 4 | **M3** | 35 | 87 | 136 | 103 | 1.19 | 337 |
| 5 | **M4** | 40 | 78 | 133 | 107 | 1.18 | 352 |
| 6 | **M5** | 20 | 70 | 100 | 110 | 1.11 | 355 |
| 7 | **M6** | 30 | 89 | 124 | 208 | 1.05 | 352 |
| 8[f] | **M7** | 24 h | 0 | - | - | - | - |
| 9 | **M8** | 30 | 93 | 151 | 103 | 1.34 | 331 |
| 10 | **M9** | 30 | 92 | 159 | 107 | 1.13 | 328 |
| 11 | **M10** | 30 | 93 | 173 | 116 | 1.49 | 280 |
| 12 | **M11** | 40 | 95 | 204 | 129 | 1.22 | 299 |
| 13 | **M12** | 30 | 88 | 176 | 123 | 1.23 | 258 |
| 14[g] | **M13** | 35 | 63 | 160 | 116 | 1.15 | 305 |
| 15[g] | **M14** | 35 | 73 | 191 | 153 | 1.85 | 305 |
| 16[h] | **M15** | 40 | 40 | 102 | n.d.[j] | n.d.[j] | 249 |
| 17 | **M16** | 35 | 86 | 219 | 68.3 | 1.22 | 271 |
| 18[i] | **M17** | 25 | 76 | 189 | n.d.[j] | n.d.[j] | 281 |

[a]Condition: Catalyst = **Zn-1**, M = 100 mg, Concentration = 1 M, initiator (I) = p-tolylmethanol, [M]:[**Zn-1**]:[I] = 1000:1:1, solvent = THF, RT.
[b]Monomer conversion measured by $^1$H NMR of the quenched solution.
[c]Calculated based on: ([M]$_0$/[I]$_0$) × Conv.% × MW$_M$ (molecular weight of monomer) + MW$_I$ (molecular weight of initiator).
[d]Number-average molecular weight ($M_n$) and dispersity index (Đ = $M_w/M_n$), determined by size exclusion chromatography (SEC) at 40 °C in THF.
[e]Determined by TGA analysis.
[f]Concentration = 2 M.
[g]Monomer didn't completely dissolve prior polymerization; For **M14**, solvent = DCM.
[h]The resulting polymer precipitated from the reaction solution.
[i]Reaction was conducted at 60 °C in 0.5 M and the resulting polymer precipitated from the reaction solution.
[j]The resulting polymer is insoluble in common solvents, and its $M_n$ and Đ cannot be measured by SEC.

disubstituted monomer library via spirocyclic substitution. A series of spirocyclic acetal-functionalized monomers **M9** – **M17** with various spirocyclic sizes and diverse functionalities was successfully prepared via acetalization of one ketone group in 1,4-cyclohexane dione with a variety of diols prior to oxidation (Fig. 3). The ROP of **M9**–**M12** at [M]:[**Zn-1**]:[I] ratio of 1000/1/1 approached 88–95% conversions within 40 min (Table 2, entries 10–13), producing polyesters P(**M9**)–P(**M12**) with $M_n$ values of 107–129 kg mol$^{-1}$. Their polymerization performance was comparable to that of **M5**. Due to its poor solubility, **M13** attached with an additional spiro-ring on the spirocyclic framework, exhibited slightly lower conversions in comparison with **M10** and **M11** (Table 2, entry 14). 73% monomer conversion in 35 min was observed for **M14**, which was modified by installing a phenyl moiety to **M10**. The introduction of an (*S,S*)-*trans*-cyclohexane fusion to **M15** led to a drastic change in the solubility of the resulting polymer P(**M15**), which precipitated out from the reaction solution during polymerization (Table 2, entry 16). In contrast, **M16** bearing a *cis*-fused ring possessed good solubility in THF and retained a similar polymerization reactivity to **M12** (Table 2, entry 17). It should be noted that the ROP of **M17** was conducted at 60 °C in a diluted solution (0.5 M) because of its poor solubility in THF, which approached 76% conversion in 25 min (Table 2, entry 18).

Since the yttrium alkyl complex **Y-1** was reported to catalyze ROP with high efficiency[90], we also examined the **Y-1**-mediated ROP of **M9**–**M17** (Fig. 2 and Supplementary Table 1). Impressively, the ROP of these monomers with only 0.01 mol% **Y-1** loading resulted in >75%

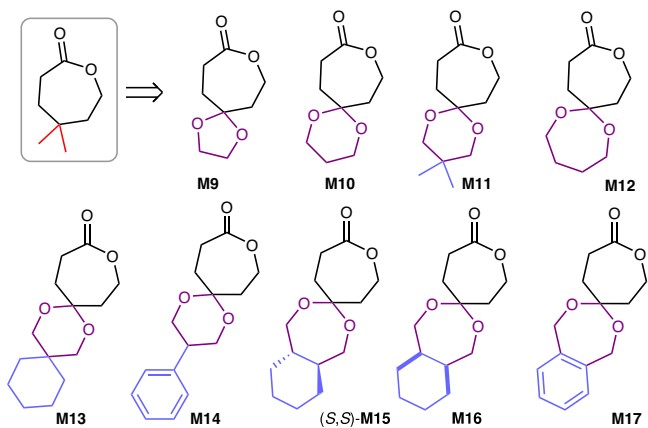

**Fig. 3 | Expanded spiro-substituted monomers.** Chemical structures of **M9**–**M17**.

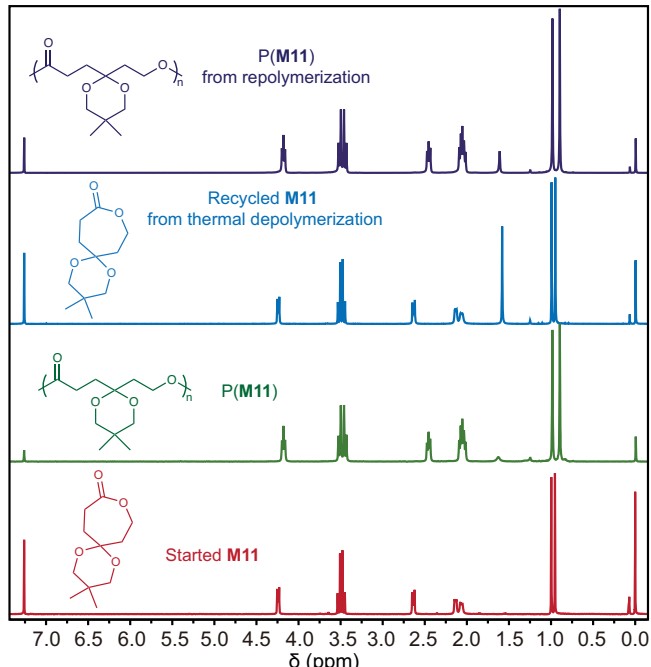

**Fig. 4 | Chemical recycling to monomer study of** M11. $^1$H NMR (CDCl$_3$, 25 °C) spectra of thermal depolymerization and repolymerization for **M11**.

conversions in 10 min (Supplementary Table 1, entries 13–15). Increasing the [**M**]:[**Y-1**]:[I] ratio to 20000:1:1 afforded P(**M11**) with an $M_{n,SEC}$ up to 494 kDa (Supplementary Table 1, entry 18). Consequently, high-molecular-weight P(**M**)s were readily synthesized on gram scales using **Y-1** for further chemical and physical property characterization.

## Chemical recycling to monomers

To evaluate the chemical recycling performance of the produced P(**M**)s, the solution depolymerization experiments were tested by mixing the polymer solution ([P(**M**)] = 20 mM in toluene based on the moles of repeat units in polymers) with 2 mol% **Zn-1** catalyst at 140 °C for 1 h. $^1$H NMR analysis of the resulting solution revealed that P(**M1**)–P(**M4**) were depolymerized back to **M1**–**M4** with conversions of 69–99% (Supplementary Table 15, entries 1–4, Supplementary Figs. 125, 128–130). In line with our thermodynamic study, increasing the substituent size of **M1**–**M4** facilitated the depolymerization process. Additionally, **M5** and **M6** were able to be recovered in 94 and 96% conversions, respectively (Supplementary Table 15, entries 5–6, Supplementary Figs. 131, 132). For the nonadjacent aromatic lactone **M8**, it showed a great monomer recovery yield of 91% compared with the other semi-aromatic monomers (Supplementary Table 15, entry 7, Supplementary Fig. 133)[75,76,91]. These results confirmed the strong correlation between depolymerizabililty and $T_c$. Among **M1**–**M8**, **M4** has the lowest $T_c$ of 241 °C and achieved nearly complete recycling (>99%), whereas **M1** with the highest $T_c$ of 814 °C, displayed only 69% recovery conversion.

The depolymerization of spirocyclic acetal-functionalized polymers P(**M9**)–P(**M12**) afforded their corresponding monomers **M9**–**M12** in increasing conversions from 88% to >99% (Supplementary Table 15, entries 8–11, Supplementary Figs. 134–137). Particularly, P(**M12**) containing a seven-membered spiro cycle demonstrated perfect chemical recyclability. These findings also provided supporting evidence that increasing the substituent size of monomers benefited the depolymerization pathway. The thermodynamic study revealed that P(**M9**)–P(**M12**) exhibited a decreasing $T_c$ tendency from 503 to 197 °C, suggesting an improvement of depolymerizability, which was consistent with our chemical recycling results. Additionally, **M13**, **M14**, and **M16** could be recovered from P(**M13**), P(**M14**), and P(**M16**) through solution depolymerization with 96, 96, and 94% conversions, respectively (Supplementary Table 15, entries 12–14 and Supplementary Figs. 138–140).

Based on our thermodynamic study, we believe these substituted CL derivatives could achieve thermal bulk depolymerization at a mild temperature. Bulk P(**M11**) (produced by [**M11**]:[**Zn-1**]:[I] ratio of 500/1/1, $M_n$ = 86.3 kg mol$^{-1}$, Đ = 1.28) with 2 mol% La[(N(SiMe$_3$)$_2$)$_3$] at 160 °C produced monomeric **M11** in 98% yield with >99% purity (Supplementary Table 17, entry 1). More importantly, the recovered **M11** was able to carry out repolymerization at [**M11**]:[**Zn-1**] ratio of 500/1

without an obvious decrease in polymerization activity, yielding the recycled P(**M11**) with an $M_n$ of 136 kg mol$^{-1}$ (Fig. 4). The improvement in $M_n$ of the resynthesized P(**M11**) was attributed to the loss of a trace amount of initiators in the system during depolymerization. Gratifyingly, P(**M15**) and P(**M17**) with poor solubility in common organic solvents were able to undergo thermal depolymerization under the above conditions and gave 95 and 92% recovery yields of their corresponding monomers **M15** and **M17**, respectively (Supplementary Table 17, entries 2 and 3). Consequently, this systematic exploration of the chemical recycling study of P(**M**)s suggested that substitution size and position were key factors for the direction of reversible polymerization and depolymerization.

## Thermal and mechanical properties

To further understand the substitution effect on the polymer properties, thermal gravimetric analysis (TGA) and differential scanning calorimetry (DSC) were employed to examine the thermal properties of the obtained P(**M**)s. The PCL derivatives P(**M1**)–P(**M8**) exhibited remarkable thermal stability with $T_d$ (onset decomposition temperature, measured by the temperature of 5% weight loss) ranging from 331 to 366 °C (Supplementary Figs. 65–71). A range of $T_d$ values from 249 to 328 °C (Fig. 5a and Supplementary Figs. 72–80) was observed for P(**M9**)–P(**M17**) containing spirocyclic acetal moieties. We hypothesized that the spirocyclic structures with high ring strain led to the decreased stability of P(**M9**)–P(**M17**).

Impressively, varying the substituents on PCL offered intriguing opportunities for tailoring the thermal properties of these PCL derivatives. P(**M1**)–P(**M8**) displayed a wide range of glass transition temperature ($T_g$) values from –67 to 18 °C (Fig. 5b) via simple modification of the rigidity of the substituents. The introduction of spirocyclic acetal functionalities rendered the resulting P(**M**)s with various $T_g$ values (–14 to 70 °C) and allowed the transformation of P(**M**)s from amorphousness to semi-crystallinity (Fig. 5c). Particularly, attaching spirocyclic structures to P(**M5**) could improve the $T_g$ values of the resulting polymers P(**M9**)–P(**M12**). It was consistent with the findings that the addition of phenyl groups or extra *trans*-fused rings to P(**M12**) led to an improvement of $T_g$ from 20 to 70 and 61 °C, respectively.

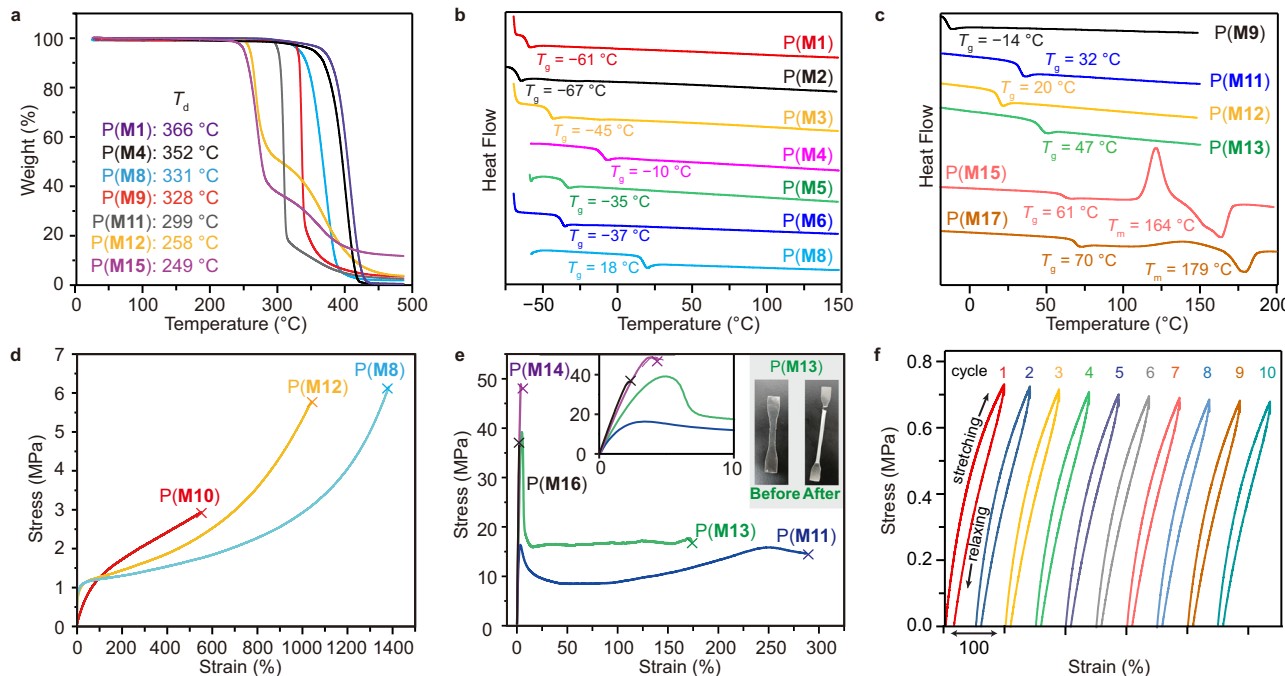

**Fig. 5 | Thermal and mechanical properties of P(M)s. a** TGA curve of representative P(M)s. **b, c** DSC curves of representative P(M)s. **d, e** Strain-stress curves of representative P(M)s. Images showing the P(M13) film. **f** Cyclic tensile testing of P(M10).

More impressively, P(M15) containing *trans*-fused rings displayed a glass transition with $T_g$ of 61 °C, a crystallization transition at 122 °C, and a melting transition with $T_m$ of 164 °C. P(M17) exhibited a $T_g$ of 70 °C and a $T_m$ of 179 °C. Collectively, a detailed analysis of structure–property relationship was delineated, highlighting the significance of the substitution effect. More importantly, a remarkable range of $T_g$ values, from −67 to 70 °C, provides the opportunity to understand the tunability of mechanical properties of the PCL-based recyclable polymers.

Tensile testing of these PCL derivatives was next investigated to gain further insight into the substitution effect on the mechanical properties of P(M)s. The P(M) specimens were prepared by melt pressing at 90−120 °C. Unfortunately, attempts to prepare P(M15) and P(M17) films failed due to their brittleness. SEC analysis of the resting P(M) films revealed that no obvious degradation was observed, indicative of their excellent thermal stability (Supplementary Table 19). These P(M) specimens were subjected to uniaxial extension experiments and exhibited distinct mechanical performance. Particularly, P(M8), P(M10), and P(M12) with $T_g$ values close to room temperature displayed thermoplastic elastomer behavior with ultimate elongation at break ($\varepsilon_B$) ranging from 585 to 1250% and tensile strength ($\sigma_B$) < 5 MPa (Fig. 5d). The elastic performance of P(M10) ($M_n = 189$ kg mol$^{-1}$, Đ = 1.61) was further assessed by 10 cyclical tensile tests where the sample was stretched to 100% strain and relaxed at a rate of 100 mm min$^{-1}$. As expected, the P(M10) sample sustained excellent elastic recovery (>90%) after ten cycles (Fig. 5f). Stress-strain curves of the P(M) films with relatively high $T_g$ values (above room temperature) were shown in Fig. 5e. P(M14) ($M_n = 241$ kg mol$^{-1}$, Đ = 1.32) containing rigid phenyl groups showed a remarkable tensile strength ($\sigma_B = 49.8 \pm 5$ MPa) with a limited strain ($\varepsilon_B = 4.1 \pm 0.3\%$) and Young's modulus ($E = 1.89 \pm 0.20$ GPa), representing a hard and brittle material. P(M13) ($M_n = 281$ kg mol$^{-1}$, Đ = 1.67) produced a yielding strength ($\sigma_Y$) of $34.7 \pm 3.6$ MPa and a breaking strength ($\sigma_B$) of $18.5 \pm 1.2$ MPa with elongation at break of $141 \pm 23\%$ and Young's modulus of $1.16 \pm 0.10$ GPa. This impressive toughness and ductility of P(M13) was comparable to the commodity plastic isotactic polypropylene[8,92]. The ductility was further improved for P(M11)

($M_n = 246$ kg mol$^{-1}$, Đ = 1.59) with $\varepsilon_B = 257 \pm 19\%$. Overall, the physical properties of P(M)s were shown to rely on the functionalities of the monomers. A simple modification of the monomer substituents could be a powerful tool to tune the chemical and physical properties of the produced material.

## Discussion

Monomer design was an important strategy to tune the polymerization thermodynamics and achieve chemical recycling to monomer. A systematic investigation was performed for the high-$T_c$ PCL system to evaluate the substitution effect and structure–property relationships. A series of substituted caprolactones were prepared to probe the change of their thermodynamic parameters for ring-opening polymerization. Increasing the steric bulk of the substitution in the CL system was proved to promote the depolymerization pathway and reduce the $T_c$ values for the system from 2060 to 241 °C. Moreover, the substituent location was also proved to have an influence on polymerization thermodynamics. The detailed substitution effect of CL derivatives (M1−M8) on polymerization thermodynamics were established to guide the future monomer design with predicted $T_c$ values.

Taking advantage of the geminal disubstituted effect, a spirocyclic substitution strategy was applied to expand the library of geminal disubstituted caprolactone-based monomers. Notably, this class of monomers (M9−M17) inherited the efficient polymerizability and excellent chemical recyclability from geminal dimethyl-substituted monomer (M5). More impressively, the spirocyclic substitution strategy imparted the resulting polymers with tunable properties by the observation of thermal transformation from amorphous to semicrystalline and mechanical transformation from brittleness to ductility, which will be vital for optimizing their performance in future applications from elastomers to plastics. This comprehensive characterization of structure–property relationships could be exploited to build a practical database for the modification and prediction of new material properties. Overall, this systematic study provided a guideline to the future monomer design towards chemically recyclable polymers and served as a toolbox for fine-tuning the material properties via functionalization.

## Methods

All synthesis and manipulations of air- and moisture-sensitive chemicals and materials were carried out in flamed Schlenk-type glassware on a dual-manifold Schlenk line, on a high-vacuum line, or in an inert gas (Ar)-filled glovebox. High-performance liquid chromatography (HPLC)-grade anhydrous tetrahydrofuran (THF), toluene (TOL), and dichloromethane (DCM) were dried via a Vigor YJC-5 solvent purification system and stored over activated Davison 4 Å molecular sieves in the glovebox. The initiator p-tolylmethanol was purchased from Adamas and purified via sublimation at 55 °C under vacuum. The other regents from Adamas-beta, Energy Chemical, and LeYan were used as received unless otherwise stated. All solid monomers were recrystallized once from DCM and petroleum ether (PE) to get the crystals of monomers. The crystals were further purified by sublimation at 90–130 °C under vacuum, and the liquid monomers were further purified via distillation at 95–130 °C/0.6–1 torr from $CaH_2$ under vacuo.

### General Procedure for the ring-opening polymerization

Polymerization reactions were performed in 4 mL glass vials inside the glovebox for ambient temperature runs. In a typical polymerization reaction, the solution of the catalyst in THF was added to the vigorously stirred prepared monomer and initiator (p-tolylmethanol) solution (THF). After a desired period of time, the polymerization was quenched by the addition of 1 mL THF acidified with benzoic acid (2%). The quenched mixture precipitated into 50 mL of cold methanol, filtered, and washed with cold methanol. This procedure was repeated twice to ensure any catalyst residue or unreacted monomer was removed. The polymer was dried in a vacuum oven at 100 °C to a constant weight.

### General procedure for the CRM of polymers in dilute solutions

A pressure tube containing the purified polymer sample (20 mg) with 2 mol% **Zn-1** in toluene (0.02 M) was sealed and heated to 140 °C (bath temperature) for 1 h under an argon atmosphere. After cooling back to room temperature, the reaction mixture was concentrated (evaporation in the watch glass or under vacuum) to give a colorless product, which was used for [1]H NMR analysis to determine the recycled monomer yield.

### Nuclear magnetic resonance (NMR)

[1]H and [13]C NMR spectra were recorded on an Agilent 400-MR DD2 or a Bruker Advance 400 spectrometer ([1]H: 400 MHz, [13]C: 100 MHz). Chemical shifts ($\delta$) for [1]H and [13]C NMR spectra are given in ppm relative to TMS. The residual solvent signals were used as references for [1]H and [13]C NMR spectra and the chemical shifts were converted to the TMS scale ($CDCl_3$: $\delta H = 7.26$ ppm, $\delta C = 77.00$ ppm). The following abbreviations were used to explain the multiplicities: s = singlet, d = doublet, t = triplet, q = quartet, and m = multiplet.

### Size exclusion chromatography (SEC)

Measurements of polymer number-average molecular weight ($M_n$) and molecular weight distributions or polydispersity index ($Đ = M_w/M_n$) were performed via size exclusion chromatography (SEC). The SEC instrument consisted of an Agilent LC system equipped with one guard column and two PL gel 5 μm mixed-C gel permeation columns and coupled with an Agilent G7162A 1260 Infinity II RI detector; The analysis was performed at 40 °C using THF as the eluent at a flow rate of 1.0 mL/min. The instrument was calibrated with nine polystyrene standards, and chromatograms were processed with Agilent OpenLab CDS Acquisition 2.5 molecular weight characterization software.

### Differential scanning calorimetry (DSC)

Melting-transition temperature ($T_m$) and glass transition temperature ($T_g$) of purified and thoroughly dried polymer samples were measured by differential scanning calorimetry (DSC) on a TRIOS DSC25, TA Instrument. All $T_g$ values were obtained from a second scan after the thermal history was removed from the first scan. The heating rate was 10 °C min and the cooling rate was 10 °C /min.

### Thermo-gravimetric analysis (TGA)

Decomposition onset temperatures ($T_{onset}$) and maximum rate decomposition temperatures ($T_{max}$) of the polymers were measured by thermal gravimetric analysis (TGA) on a TGA55 Analyzer, TA Instrument. Polymer samples were heated from ambient temperatures to 500 °C at a heating rate of 10 °C/min. Values of $T_{max}$ were obtained from derivative (wt%/°C) vs. temperature (°C) plots and defined by the peak values, while $T_{onset}$ values were obtained from wt% vs. temperature (°C) plots and defined by the temperature of 5% weight loss.

### Mechanical analysis

Tensile stress/strain testing was performed by an Instron 34SC-1 universal testing system at ambient temperature. Samples were made by hot pressure in steel molds ($50 \times 4 \times 0.4$ or $50 \times 4 \times 0.8$ mm³) at 100−150 °C. The dog-bone-shaped test specimens (ca. 0.4–0.8 mm (thickness) × 4 mm (width) × 20 mm (grip width)) were stretched at a strain rate of 20−100 mm/min until break. The measurements were performed with five to ten replicates per material to report average values and standard deviations for each set. Young's modulus was calculated using the slope of the stress-strain curve from 0 to 1% strain. All samples were tested at ambient temperature (-25 °C).

### Matrix-assisted laser desorption/Ionization time-of-flight mass spectroscopy (MALDI−TOF MS)

An AXIMA performance instrument was used in reflection mode with Dithranol as the matrix. A thin layer of a NaI solution (1 μL, 0.01 mmol/mL in THF) was first deposited on the target plate, followed by the solutions of the matrix (2 μL, 10 mg/mL in $CHCl_3$) and polymer (2 μL, 5 mg/mL in THF) were mixed. The mixed solution was spotted on the MALDI sample plate and air-dried. The raw data was processed in the Shimadzu Biotech MALDI-MS software.

## Data availability

The authors declare that the data supporting the findings of this study are provided in the main article and the Supplementary Information. All data is available from the corresponding author upon request.

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

## Acknowledgements

This work was supported by the National Key R&D Program of China (2021YFA1501700), the National Natural Science Foundation of China (22071163 and U19A2095), the "1000-Youth Talents Program", and the Fundamental Research Funds for the Central Universities (YJ201924 and YJ202209). We would like to thank Dr. Dongyan Deng from the College of Chemistry, Sichuan University, for NMR testing. We also would like to thank Prof. Peiyuan Yu from the Southern University of Science and Technology for the polymerization thermodynamics discussion.

## Author contributions

J.-B.Z. conceived the project. J.-B.Z. and Z.C. directed the research. Y.-M.T. and F.-L.G. designed and performed experiments related to monomer and polymer synthesis. Y.-M.T. designed and performed the characterization experiments. Y.-M.T. and Y.-C.W. performed the depolymerization experiments. Z.C. and Y.-M.T. wrote the initial manuscript draft, J.-B.Z. edited the draft, and all the authors contributed to the revised manuscript.

## Competing interests

The authors declare no competing interests.
