## [Peer Review File · Nature Communications]

Insights into Substitution Strategy towards Thermodynamic and Property Regulation of Chemically Recyclable PolymersReviewers' Comments:

Reviewer #1:

Remarks to the Author:

In their manuscript, Tu et al. showed the substituent effects in a series of ϵ -caprolactone based monomers in the context of chemically recyclable polymers. The substituents can impact the thermodynamics in polymerization and the thermomechanical properties of the materials. The structure-reactivity relationship in the polymerization of cyclic monomers was also studied in detail. This study is a timely one that fits well into the important field of chemically recyclable polymers. I recommend its publication in Nature Communications after the authors address the following concerns:

1. The authors note that the substituent size has a significant effect on the polymerizability of the monomers. Particularly, for monomers M1-M4 they note that there is a decrease in enthalpy (ΔH_p) and an increase in entropy (ΔS_p) of polymerization that appears to correspond to the increasing substituent size, leading to the overall decrease in polymerizability. The overall discussion around polymerizability would be stronger if the authors could further rationalize why these substituents affect ΔH_p and ΔS_p the way they do. A rationalization of such behavior from the molecular level would be helpful.
2. The authors use TOF values to compare the polymerization activity of unsubstituted and substituted ϵ -caprolactones. While it appears that these were calculated using the polymerization conversion and reaction times mentioned in Table 2, this has not been explicitly mentioned in the main text or SI. If the TOFs are calculated at a reaction time that is significantly beyond the time required to reach maximum conversion, it becomes difficult to compare TOFs for different monomers to assess their polymerization activity or kinetics. This is further compounded by the fact that a monomer with a lower T_c may have a lower equilibrium monomer conversion at the given conditions, resulting in a lower apparent TOF even if the polymerization actually reaches its equilibrium conversion faster.
3. Table 1 and Table S2: The standard deviation or error associated with the thermodynamic properties has not been reported. These should be included to better facilitate the comparison of these quantities for different monomers.
4. On page 119 of the SI, the citation of Figure S116 should be Figure S115.
5. Why is chain transfer more significant in other substrates than in M1-M5?
6. The thermodynamics data for monomers M9-M12 consists of only a single replicate each. Performing these studies in triplicate similar to monomer M1-M8 would make the comparison between these two groups more reliable.
7. Chemical recycling was conducted at 140 °C in toluene, 30 °C higher than the boiling point of toluene. Is 140 °C the temperature of the bath?
8. The authors should indicate which NMR peaks were used to calculate the extent of depolymerization in the chemical recycling studies.
9. Some polymers show a significant endotherm during the first heating cycle of DSC experiments. Particularly, polymers P(M8), P(M12), P(M14) (Fig. S83, S87 and S89). Do the authors have any hypotheses as to why these endotherms appear on the first heating cycle but not subsequent heating cycles?
10. In Figure 4, the authors use wedged bonds to show trans stereochemistry (M15) but use bold bonds to present cis (M16). According to the IUPAC recommendation, wedged bonds should be used to indicate stereochemistry.

Reviewer #2:

Remarks to the Author:

The paper purports to offer a unique insight into the structure property relationships for a range of cyclic esters, namely substitute caprolactones. There is an extensive amount of work showcased in the paper, but much of it is iterative, and builds on even more extensive literature precedent (many of

which are uncited, either missing follow-on papers from single publications cited, or key papers completely missed by the authors who seem to focus their reviews on specific authors known to their team). I'll decline to list the ones that our team has published, as I'm not grasping for more citations – but this needs to be improved.

Not only would additional papers need citing, but they need to be contextualised relative to previous work. There is extensive knowledge both about substitution patterns in polyesters (there's nothing special about caprolactones – contextualise vs the plethora of other polyesters) and in the impact of topology, catalyst and conditions on ceiling temperatures which need to be brought into an effective discussion to showcase what is actually new here rather than just an extensive and thorough body of work. There is a tendency to simply present data rather than discuss its relevance relative to the state of the art.

The introduction is somewhat overzealous – polymer waste is indeed a serious environmental issue, but this waste isn't arising from caprolactones. Much of the waste needs a fate like mechanical recycling or a broad scope chemical recycling technology like hydrocracking. Contextualising this in the broader field is important, but the authors focus their introduction on thermodynamics without any understanding of sustainability. If this is to be a circular economy, and Nature Communications a broad scope journal, what is the proposed evaluation of sustainability from a metrics perspective that would justify publication? The novelty is disclosed in the authors previous works, predominantly, and so the systemic issues are what will make this paper impactful.

The work is focussed on lowering the T_c value for caprolactones. This will, of course, lower the commercial relevance of the PCLs themselves, as they are intricately linked, and indeed add significantly to the synthetic carbon footprint due to the complexity of the monomers. This seems to be disregarded, as is much of the previous work on substituted caprolactones.

The choice of catalysts is esoteric, and while the conclusions drawn are focussed on the substituent effects of the monomers, bulky ligands and indeed bulky amides that require pre-initiation. Immediate questions of the impact of these conditions on the work spring to mind, as do questions around why more commonplace and less expensive organocatalysts or something like $\text{Sn}(\text{Oct})_2$ or Al-salen systems were not explored, especially as these have been used on substituted caprolactones before (sometimes in missing papers that need citing).

What then transpires is a lot of work (and I mean a lot of work), with very little systematic discussion and almost no contextualisation with what has already been published in the literature. The authors seem to only have cited "chemical recycling" references rather than exploring the literature for substituted caprolactones or other 7-membered rings nor drawn conclusions from other substitution patterns in other polyesters. The results, when looked at through the lens of previous papers, aren't really that surprising – of course you can lower the T_c with these substitutions and tune the thermal properties, etc.

The experimental is poor. There appears to be many formatting errors, the ChemDraws are inconsistent and many are missing symbols, and the characterisation is poor for both small molecule products (^1H NMR spectra are not enough to show purity or prove composition – where are the other NMR experiments and either a HR-MS or elemental analysis?). When there is a ^{13}C spectrum, it is labelled as a " ^{13}H NMR" such as in Figure S34! Did the authors not check this file after uploading? This poor quality suggests a lack of attention to detail from the authors.

There also is a glaring lack of information on reaction kinetics and progress – conclusions are drawn from timepoints when known reactivity ratios and substitution patterns will suggest very different initiation and propagation rates, but no kinetics (or Mn vs conversion plots) are provided in either the main manuscript or the SI. The Van't Hoff plot is fine, but this isn't anything but a snapshot. The rest of the polymer characterisation is also lacking, with plots inconsistent in the DSC cycling (i.e. heat,

cool, heat; lack of definition in peaks, spectra appear pasted in to generic scale diagram. Better, more consistent spectra, more clearly showing the desired features would be beneficial. The same could be said for the SEC, which again look copy/pasted in. Everything is repetitive and poorly presented, almost in an effort to make it longer rather than informative. This isn't a thesis, but a curated paper submitted to an exceptionally high impact journal.

In all, this is not an appropriate paper for Nature Communications, as while the volume of work is very high it is written almost exclusively for a polymer audience, it doesn't understand sustainability challenges in polymer recycling, it doesn't contextualise the results versus the literature, and instead simply presents masses of data in a repetitive way without showing new insights or impact. I would suggest the authors use the comments to revise the manuscript into a strong Macromolecules or Polymer Chemistry paper, as that is the appropriate audience for this work.

Reviewer #3:

Remarks to the Author:

The manuscript applied substitution strategy to tune the thermodynamics of seven-membered ring caprolactone derivatives. The authors designed and prepared a series of alkyl substituted and spirocyclic acetal-functionalized caprolactones and examined their ROP behaviors by using zinc and yttrium based catalysts. They systematically studied the ROP thermodynamic of obtained cyclic monomers and demonstrated that the chemical recycling properties of resultant polymers depended on the size, position and structures of their respective substituents. Their finding demonstrated that such strategy can offer various opportunity to prepare circular close-loop polymer with high melting temperature and mechanical properties. This represented a systematic strategy to create a serial close-loop polyesters with tunable melting temperature and potential high performances polymers. These finding contributed important advances in the field of close-loop polymers, and it deserves publications in Nature Comm. Moreover, the manuscript is well-written and the conclusions are well supported by the experimental data. Thus, I recommend accept after some minor revisions outlined below:

1. The symbol of Gibbs free energy should be italicized in Figure 1. I recommend that the values of T_c should be added in Figure 1a, that the effects of different fused-ring structure on T_c s can be compared more clearly.
2. In page 5 line 96, Table 1 and footnote of Table 2, the format of initial monomer concentration (like 1.0 M) was different from others (1M) in manuscript. In page 7 line 129, "83.0–137 kg mol⁻¹" also should be revised. In page 7 line 145, "M12" should be revised into "M15". In Figure 4, the "Repolymerization" at first curve should be revised into "repolymerization". In page 14 line 299, the type size should be revised. The abbreviation of reference 55 should be revised.
3. Monomer M7 with geminal dimethyl substitutions on both β and δ positions cannot polymerize at RT. Did the author try different catalysts or ROP conditions such as low temperature? At least, decrease ROP temperature should be attempted.
4. The M_n ,SEC of PM6 is obviously lower than theoretical value, while the distribution remains narrow (1.14). It may be ascribed to the multiple initiation caused by impurities rather than chain transfer. Corresponding MALDI-TOF MS of PM6 should be provided to further demonstrate the results.
5. According to the thermodynamic formula, the T_c of M1 is calculated as 70 °C at 0.02M, which is lower than 140 °C. Hence, PM1 should be complete degraded at 140 °C in 0.02M. However, the depolymerization of PM1 only reached 69%. And Figure S128 showed obvious signals of side products that did not belong to recovered monomers and polymers. Hence, it may be a factor affecting the conversion. What is the structures of side products? Can the by-products be reduced by conducting the depolymerization at lower temperatures? Also, have the authors tried different degradation concentration?
6. The bp of toluene is about 110 °C, but the depolymerization reaction in toluene was performed at 140 °C. Please give more details in experiments of how to realize this.
7. In page 8 line 164, I want to know whether the "[P(M)] = 20 mM in toluene" and "2 mol % Zn-1"

was based on moles of polymers or structural units in polymers. In addition, the conversion rather than yield can be calculated according to in-situ ^1H NMR spectra.

8. The DSC curve of PM5 should be added to Figure b or c.

9. Did the polymers (PM8, PM10 and PM12) possess stiffness to undergo mechanical testing at 25 °C, because their T_g s are lower than 25 °C? It is interesting that these polymers with low T_g s have good mechanical properties, lack of crystallization behavior or crosslink structures.

Changes Made in the Revision and Responses to the Comments by the Reviewers

Reviewer #1 (Remarks to the Author):

Reviewer's general comment: In their manuscript, Tu et al. showed the substituent effects in a series of ϵ -caprolactone based monomers in the context of chemically recyclable polymers. The substituents can impact the thermodynamics in polymerization and the thermomechanical properties of the materials. The structure-reactivity relationship in the polymerization of cyclic monomers was also studied in detail. This study is a timely one that fits well into the important field of chemically recyclable polymers. I recommend its publication in Nature Communications after the authors address the following concerns:

Our response: Thank you very much for your kind comments and recommendation for publication in *Nat. Commun.*. We also thank your valuable comments that helped us improve the quality of the manuscript. We have revised the manuscript according to your specific comments listed below.

Reviewer's specific comments:

1. The authors note that the substituent size has a significant effect on the polymerizability of the monomers. Particularly, for monomers M1-M4 they note that there is a decrease in enthalpy (ΔH_p) and an increase in entropy (ΔS_p) of polymerization that appears to correspond to the increasing substituent size, leading to the overall decrease in polymerizability. The overall discussion around polymerizability would be stronger if the authors could further rationalize why these substituents affect ΔH_p and ΔS_p the way they do. A rationalization of such behavior from the molecular level would be helpful.

Our response: We appreciate the reviewer for this constructive suggestion and have added more discussion in the manuscript. The value of ΔH_p directly reflected the ring strain of monomer. A simple increase in substituent size from methyl to tert-butyl group led to a slight decrease in ring strain of corresponding monomers due to the increasing stabilization of ring structure. In contrast, substituent numbers and positions that could contribute to more conformation and repulsion strain appeared to have more pronounced influence on the monomeric ring strain and facilitate a large increase in the value of ΔH_p . Meanwhile, substitution on CL improve the magnitude of the "increased order of the system" significantly, which represented the significant increase in the value of ΔS_p . Naturally, substituent numbers and positions could devote more "disorder" to the system as well.

"In comparison to substituent size, substituent numbers and positions that could contribute to more conformation and repulsion strain appeared to have more pronounced influence on the monomeric ring strain and facilitate a large increase in the value of ΔH_p . Substitution on CL could devote significant "disorder" to the system, leading to the increase in the value of ΔS_p ."

2. The authors use TOF values to compare the polymerization activity of unsubstituted and substituted ϵ -caprolactones. While it appears that these were calculated using the polymerization conversion and reaction times mentioned in Table 2, this has not been explicitly mentioned in the main text or SI. If the TOFs are calculated at a reaction time that is significantly beyond the time required to reach maximum conversion, it becomes difficult to compare TOFs for different monomers to assess their polymerization activity or kinetics. This is further compounded by the fact that a monomer with a lower T_c may have a lower equilibrium monomer conversion at the given conditions, resulting in a lower apparent TOF even if the polymerization actually reaches its equilibrium conversion faster.

Our response: We would like to thank the reviewer for this helpful insight. We have probed the polymerization kinetics of **M1–M8** by ^1H NMR spectroscopy. The conversion vs. time plots were summarized in Supplementary Figure 30. These monomers approached polymerization equilibrium within 40 mins at room temperature with a $[\text{M}]:[\text{Zn-1}]:[\text{I}]$ ratio of 1000:1:1. Similar polymerization activity were observed for **M1–M8**. Based on the reviewer's comment and the above kinetic results, we also considered that TOF values we calculated may be misleading. Consequently, we have removed our statement related to TOFs in the main text and Table 2.

When we revisited these polymerization reactions, it's found that **M6** with an improved purity exhibited a similar polymerization reactivity with other monomers, approach 89% monomer conversion within 30 min. We have updated entry 7 in Table 2.

Supplementary Figure 30 was copied below.

Supplementary Figure 30 The conversion vs. time plots of **M1–M8**.

3. Table 1 and Table S2: The standard deviation or error associated with the thermodynamic properties has not been reported. These should be included to better facilitate the comparison of these quantities for different monomers.

Our response: We appreciate the reviewer's suggestion and have provided the standard deviation associated with the thermodynamic properties in Table 1.

Updated Table 1 were copied below.

Table 1. Polymerization thermodynamic data for **M1–M8**.

Entry	M	ΔH_p° (kJ mol ⁻¹)	ΔS_p° (J mol ⁻¹ K ⁻¹)	T_c at 1 M (°C)
1 ^[a]	CL	-14.0	-6.0	2060
2	M1	-16.2 ± 1.16	-14.9 ± 3.58	814
3	M2	-14.0 ± 0.50	-14.0 ± 1.50	727
4	M3	-14.8 ± 0.67	-21.3 ± 2.00	422
5	M4	-12.7 ± 0.75	-24.7 ± 2.00	241
6	M5	-15.6 ± 0.42	-22.4 ± 1.16	423
7	M6	-20.0 ± 1.50	-34.4 ± 4.57	308
8	M8	-17.5 ± 0.42	-23.2 ± 1.33	481

[a] Thermodynamic data was reported in reference 37.

For Supplementary Table 2, the thermodynamics of **M9–M12** was performed by NMR-tube experiments and one set of data was collected.

4. On page 119 of the SI, the citation of Figure S116 should be Figure S115.

Our response: Correction has been made.

5. Why is chain transfer more significant in other substrates than in M1-M5?

Our response: We believe that the purity of each monomer we synthesized could be slightly different even with their purity > 99%. The presence of impurity in the polymerization systems could contribute to more significant chain transfer and more undesired initiation.

We have revised our statement. "These $M_{n,SEC}$ values of P(M)s were inconsistent with their corresponding theoretical values calculated from $[M]_0/[I]_0$ ratios and conversions ($M_{n,Calcd}$), demonstrating a *not* well-controlled polymerization system, since the high reactivity of these monomers led to undesired initiation events and inevitable chain transfer."

6. The thermodynamics data for monomers M9-M12 consists of only a single replicate each. Performing these studies in triplicate similar to monomer M1-M8 would make the comparison between these two groups more reliable.

Our response: We understand the reviewer's concern. For the thermodynamic study of **M9–M12**, we used the variable temperature NMR spectroscopy to monitor the polymerization since these monomers were prone to acetal hydrolysis under the acidic conditions. In comparison to the thermodynamic study of **M1–M8** which was set on three parallel samples at each temperature and quenched to determine the monomer concentration at equilibrium, we believe the NMR-tube experiment was more reliable, which directly monitor the polymerization equilibrium at each temperature and ensured the reversibility of the polymerization equilibrium to minimize the system errors and improve the reproducibility.

7. Chemical recycling was conducted at 140 °C in toluene, 30 °C higher than the boiling point of toluene. Is 140 °C the temperature of the bath?

Our response: We apologize for this confusing description. To clarify, 140 °C was the temperature of the bath and a pressure tube was used for these chemical recycling experiments.

We have provided these details in the supporting information.

“A pressure tube containing the purified polymer sample (20 mg) with 2 mol% Zn-1 in toluene (0.02 M) was sealed and heated to 140 °C (bath temperature) for 1 h under an argon atmosphere. After cooling back to room temperature, the reaction mixture was concentrated (evaporation in the watch glass or under vacuum) to give a colorless product, which was used for ¹H NMR analysis to determine the recycled monomer conversion.”

8. The authors should indicate which NMR peaks were used to calculate the extent of depolymerization in the chemical recycling studies.

Our response: We have added the new zoomed-in figures in the supporting informaton to illustrate how we calculate the extent of depolymerization in the chemical recycling studies as the reviewer suggested.

Some representative examples are copied below.

Supplementary Figure 1 ¹H NMR spectra of a) recycled **M1** by the solution depolymerization (Monomer conversion = $3.00/(3.00 + 1.35) = 0.69$), top; b) starting **M1** for comparison, middle; c) P(**M1**) ($M_n = 81.9$ kg/mol, $D = 1.68$), bottom.

Supplementary Figure 2 ¹H NMR spectra of a) recycled **M2** by the solution depolymerization (Monomer conversion = $3 \times 1.00/3.40 = 0.88$), top; b) starting **M2** for comparison, middle; c) P(**M2**) ($M_n = 158$ kg/mol, $D = 1.48$), bottom.

Supplementary Figure 3 ^1H NMR spectra of a) recycled **M4** by the solution depolymerization (Monomer conversion = $9 \times 1.00 / 9.05 = 0.99$), top; b) starting **M4** for comparison, middle; c) P(**M4**) ($M_n = 42.9$ kg/mol, $D = 1.36$), bottom.

Supplementary Figure 4 ^1H NMR spectra of a) recycled **M5** by the solution depolymerization (Monomer conversion = $6.00 / (6.00 + 0.36) = 0.94$), top; b) starting **M5** for comparison, middle; c) P(**M5**) ($M_n = 60.4$ kg/mol, $D = 1.66$), bottom.

Supplementary Figure 5 ¹H NMR spectra of a) recycled **M6** by the solution depolymerization (Monomer conversion = $2.00 / (2.00 + 0.08) = 0.96$), top; b) starting **M6** for comparison, middle; c) **P(M6)** ($M_n = 83.0$ kg/mol, $\bar{D} = 1.14$), bottom.

9. Some polymers show a significant endotherm during the first heating cycle of DSC experiments. Particularly, polymers **P(M8)**, **P(M12)**, **P(M14)** (Fig. S83, S87 and S89). Do the authors have any hypotheses as to why these endotherms appear on the first heating cycle but not subsequent heating cycles?

Our response: We believe this phenomenon is attributed to the slow crystallization rate of these semi-crystalline polymers **P(M8)**, **P(M12)**, and **P(M14)**. Upon removal of the thermal history in the first heating scan, the controlled cooling scan and the second heating scan give these samples a new thermal history, but these samples didn't get proper conditions for crystallization during the process. That's why these endotherms appear on the first heating cycle but not subsequent heating cycles.

10. In Figure 4, the authors use wedged bonds to show trans stereochemistry (**M15**) but use bold bonds to present cis (**M16**). According to the IUPAC recommendation, wedged bonds should be used to indicate stereochemistry.

Our response: Based on the reviewer's question, we believe the reviewer referred to Fig 3 in the manuscript. We apologize for our confusing description in Fig 3 and have revised Fig 3. Here we would like to emphasize *cis*-**M16** is a *rac*-monomer containing a mixture of (*R,S*)-**M16** and (*S,R*)-**M16**. In comparison, *trans*-**M15** is enantiopure (*S,S*)-**M15**.

The revised Fig 3 was copied below.

Fig. 3 Expanded spiro-substituted monomers.

Reviewer #2 (Remarks to the Author):

Reviewer's general comment: The paper purports to offer a unique insight into the structure property relationships for a range of cyclic esters, namely substitute caprolactones. There is an extensive amount of work showcased in the paper, but much of it is iterative, and builds on even more extensive literature precedent (many of which are uncited, either missing follow-on papers from single publications cited, or key papers completely missed by the authors who seem to focus their reviews on specific authors known to their team). I'll decline to list the ones that our team has published, as I'm not grasping for more citations – but this needs to be improved.

Our response: We sincerely welcome and appreciate these comments from the reviewer. We revised the manuscript according to your specific comments listed below. In this new version of manuscript, we would like to show that we added more related references.

References:

- 56 Urakami, H. & Guan, Z. Living ring-opening polymerization of a carbohydrate-derived lactone for the synthesis of protein-resistant biomaterials. *Biomacromolecules* **9**, 592-597, (2008).
- 57 Peeters, J. *et al.* Cascade synthesis of chiral block copolymers combining lipase catalyzed ring opening polymerization and atom transfer radical polymerization. *Biomacromolecules* **5**, 1862-1868, (2004).
- 58 Trollsås, M. *et al.* Highly Functional Branched and Dendri-Graft Aliphatic Polyesters through Ring Opening Polymerization. *Macromolecules* **31**, 2756-2763, (1998).
- 59 Palard, I., Schappacher, M., Soum, A. & Guillaume, S. M. Ring-opening polymerization of ϵ -caprolactone initiated by rare earth alkoxides and borohydrides: a comparative study. *Polym. Int.* **55**, 1132-1137, (2006).
- 60 El Habnoui, S. *et al.* MRI-visible poly(epsilon-caprolactone) with controlled contrast agent ratios for enhanced visualization in temporary imaging applications. *Biomacromolecules* **14**, 3626-3634, (2013).
- 61 Jazkewitsch, O., Mondrzyk, A., Staffel, R. & Ritter, H. Cyclodextrin-Modified Polyesters from Lactones and from Bacteria: An Approach to New Drug Carrier Systems. *Macromolecules* **44**, 1365-1371, (2011).
- 62 Li, H., Jérôme, R. & Lecomte, P. Synthesis of tadpole-shaped copolyesters based on living macrocyclic poly(ϵ -caprolactone). *Polymer* **47**, 8406-8413, (2006).
- 63 Tian, D., Dubois, P., Grandfils, C. & Jérôme, R. Ring-Opening Polymerization of 1,4,8-Trioxaspiro[4.6]-9-undecanone: A New Route to Aliphatic Polyesters Bearing Functional Pendent Groups. *Macromolecules* **30**, 406-409, (1997).
- 64 Mecerreyes, D., Trollsås, M. & Hedrick, J. L. ABC BCD Polymerization: A Self-Condensing Vinyl and Cyclic Ester Polymerization by Combination Free-Radical and Ring-Opening Techniques. *Macromolecules* **32**, 8753-8759, (1999).

- 65 Liu, M., Vladimirov, N. & Fréchet, J. M. J. A New Approach to Hyperbranched Polymers by Ring-Opening Polymerization of an AB Monomer: 4-(2-Hydroxyethyl)- ϵ -caprolactone. *Macromolecules* **32**, 6881-6884, (1999).
- 66 Lou, X., Detrembleur, C., Lecomte, P. & Jérôme, R. Living Ring-Opening (Co)polymerization of 6,7-Dihydro-2(5H)-oxepinone into Unsaturated Aliphatic Polyesters. *Macromolecules* **34**, 5806-5811, (2001).
- 67 Tan, L., Maji, S., Mattheis, C., Chen, Y. & Agarwal, S. Antimicrobial hydantoin-grafted poly(epsilon-caprolactone) by ring-opening polymerization and click chemistry. *Macromol. Biosci.* **12**, 1721-1730, (2012).
- 68 Mecerreyes, D., Miller, R. D., Hedrick, J. L., Detrembleur, C. & Jérôme, R. Ring-opening polymerization of 6-hydroxynon-8-enoic acid lactone: Novel biodegradable copolymers containing allyl pendent groups. *J. Polym. Sci. A Polym. Chem.* **38**, 870-875, (2000).
- 69 Massoumi, B., Abdollahi, M., Fathi, M., Entezami, A. A. & Hamidi, S. Synthesis of novel thermoresponsive micelles by graft copolymerization of N-isopropylacrylamide on poly(ϵ -caprolactone-co- α -bromo- ϵ -caprolactone) as macroinitiator via ATRP. *J. Polym. Res.* **20**, (2013).
- 70 Tran, H. *et al.* Toward Recyclable Polymers: Ring-Opening Polymerization Enthalpy from First-Principles. *J. Phys. Chem. Lett.* **13**, 4778-4785, (2022).
- 71 Malikmammadov, E., Tanir, T. E., Kiziltay, A., Hasirci, V. & Hasirci, N. PCL and PCL-based materials in biomedical applications. *J. Biomater. Sci. Polym. Ed.* **29**, 863-893, (2018).
- 72 Ciani, L. *et al.* Rational design of gold nanoparticles functionalized with carboranes for application in Boron Neutron Capture Therapy. *Int. J. Pharm.* **458**, 340-346, (2013).
- 73 Bartnikowski, M., Dargaville, T. R., Ivanovski, S. & Hutmacher, D. W. Degradation mechanisms of polycaprolactone in the context of chemistry, geometry and environment. *Prog. Polym. Sci.* **96**, 1-20, (2019).
- 74 Li, H., Debuigne, A., Jérôme, R. & Lecomte, P. Synthesis of macrocyclic poly(epsilon-caprolactone) by intramolecular cross-linking of unsaturated end groups of chains precyclic by the initiation. *Angew. Chem. Int. Ed.* **45**, 2264-2267, (2006).
- 75 Su, R.-J., Yang, H.-W., Leu, Y.-L., Hua, M.-Y. & Lee, R.-S. Synthesis and characterization of amphiphilic functional polyesters by ring-opening polymerization and click reaction. *React. Funct. Polym.* **72**, 36-44, (2012).

Not only would additional papers need citing, but they need to be contextualised relative to previous work. There is extensive knowledge both about substitution patterns in polyesters (there's nothing special about caprolactones – contextualise vs the plethora of other polyesters) and in the impact of topology, catalyst and conditions on ceiling temperatures which need to be brought into an effective discussion to showcase what is actually new here rather than just an

extensive and thorough body of work. There is a tendency to simply present data rather than discuss its relevance relative to the state of the art.

Our response: We appreciate the reviewer's suggestion and have provided more discussion as the reviewer suggested. "In fact, many CL-based monomers have been reported for ROP toward functionalized polyesters.⁴⁷⁻⁷⁵ These precedents revealed that ring size and degree of substitution of monomers could affect their polymerization thermodynamic equilibrium behavior. However, the relationship between T_c and chemical recyclability was not specifically studied. The renaissance of this fundamental study would provide monomer design principle towards the development of chemically recyclable polymers."

The introduction is somewhat overzealous – polymer waste is indeed a serious environmental issue, but this waste isn't arising from caprolactones. Much of the waste needs a fate like mechanical recycling or a broad scope chemical recycling technology like hydrocracking. Contextualising this in the broader field is important, but the authors focus their introduction on thermodynamics without any understanding of sustainability. If this is to be a circular economy, and Nature Communications a broad scope journal, what is the proposed evaluation of sustainability from a metrics perspective that would justify publication? The novelty is disclosed in the authors previous works, predominantly, and so the systemic issues are what will make this paper impactful. The work is focussed on lowering the T_c value for caprolactones. This will, of course, lower the commercial relevance of the PCLs themselves, as they are intricately linked, and indeed add significantly to the synthetic carbon footprint due to the complexity of the monomers. This seems to be disregarded, as is much of the previous work on substituted caprolactones.

Our response: We understand the reviewer's concern but respectfully disagree with the reviewer on the significance of our work. Although the plastic waste that caused the serious environmental issue isn't arising from polycaprolactones, a new chemically recyclable polymer with properties that are commensurate with commercial plastics could serve as an alternative and replace the traditional nondegradable plastics to address the accumulation of plastic waste. We do realize our proof-of-concept laboratory study is still far from practical application due to the scalability, the economical consideration, and the feasibility of industrial recycling process. However, we believe this fundamental study could provide a solid support in principle for monomer design strategy and serve as a guideline to drive the development of chemically recyclable polymers. Ultimately, we have faith that a scalable, economically competitive, and chemically recyclable polymer system with properties comparable to today's plastic materials would be designed/discovered after massive efforts devoted to this research field.

The choice of catalysts is esoteric, and while the conclusions drawn are focussed on the substituent effects of the monomers, bulky ligands and indeed bulky amides that require pre-initiation. Immediate questions of the impact of these conditions on the work spring to mind, as

do questions around why more commonplace and less expensive organocatalysts or something like Sn(Oct)₂ or Al-salen systems were not explored, especially as these have been used on substituted caprolactones before (sometimes in missing papers that need citing).

Our response: These catalysts we chose in this manuscript were found to be highly effective catalysts for ring-opening polymerization reactions in previous reports.¹⁻⁸ These catalysts were also proved to an efficient catalyst for ROP of CL-based monomers in this work and large-scale polymer products with high M_n s could be successfully prepared with these catalysts.

References:

- S1. Amgoune, A., Thomas, C. M., Roisnel, T. & Carpentier, J. F. Ring-opening polymerization of lactide with group 3 metal complexes supported by dianionic alkoxy-amino-bisphenolate ligands: Combining high activity, productivity, and selectivity. *Chem. Eur. J.* **12**, 169–179 (2005).
- S2. Amgoune, A., Thomas, C. M., Ilinca, S., Roisnel, T. & Carpentier, J. F. Highly active, productive, and syndiospecific yttrium initiators for the polymerization of racemic β -butyrolactone. *Angew. Chem. Int. Ed.* **45**, 2782–2784 (2006).
- S3. Kränzlein, M., Pehl, T. M., Adams, F. & Rieger, B. Uniting Group-Transfer and Ring-Opening Polymerization—Block Copolymers from Functional Michael-Type Monomers and Lactones. *Macromolecules* **54**, 10860–10869 (2021).
- S4. Liu, X., Hua, X. & Cui, D. Copolymerization of Lactide and Cyclic Carbonate via Highly Stereoselective Catalysts to Modulate Copolymer Sequences. *Macromolecules* **51**, 930–937 (2018).
- S5. Cheng, M., Attygalle, A. B., Lobkovsky, E. B. & Coates, G. W. Single-Site Catalysts for Ring-Opening Polymerization: Synthesis of Heterotactic Poly(lactic acid) from rac-Lactide. *J. Am. Chem. Soc.* **121**, 11583–11584 (1999).
- S6. Jia, Z. *et al.* Isotactic-Alternating, Heterotactic-Alternating, and ABAA-Type Sequence-Controlled Copolyester Syntheses via Highly Stereoselective and Regioselective Ring-Opening Polymerization of Cyclic Diesters. *J. Am. Chem. Soc.* **143**, 4421–4432 (2021).
- S7. Bouyahyi, M. & Duchateau, R. Metal-Based Catalysts for Controlled Ring-Opening Polymerization of Macrolactones: High Molecular Weight and Well-Defined Copolymer Architectures. *Macromolecules* **47**, 517–524 (2014).
- S8. Kronast, A. *et al.* Electron-Deficient β -Diiminato-Zinc-Ethyl Complexes: Synthesis, Structure, and Reactivity in Ring-Opening Polymerization of Lactones. *Organometallics* **35**, 681–685 (2016).

What then transpires is a lot of work (and I mean a lot of work), with very little systematic discussion and almost no contextualisation with what has already been published in the literature. The authors seem to only have cited “chemical recycling” references rather than exploring the literature for substituted caprolactones or other 7-membered rings nor drawn conclusions from other substitution patters in other polyesters. The results, when looked at through the lens of

previous papers, aren't really that surprising – of course you can lower the T_c with these substitutions and tune the thermal properties, etc.

Our response: We have added more references as the reviewer suggested. We also agree with the reviewer that previous study have showed that substitution effects could affect the T_c of the system and the thermal and mechanical properties of the resulting polymers. These findings provided supporting evidence for our current work. However, we would like to emphasize that a systematic investigation on structure–de/polymerizability and structure–property relationships in the view of chemically recyclable polymers still remained to explore. We believe it's of significance to reevaluate the substitution strategy on polymerization thermodynamics and polymer property regulation to provide a guideline to the future monomer design towards chemically recyclable polymers. In our present work, the detailed substitution effect of CL derivatives on polymerization thermodynamics were established to guide the future monomer design with predicted T_c values. This comprehensive characterization of structure–property relationships could be exploited to build a practical database for the modification and prediction of new material properties. Overall, this systematic study provided a guideline to the future monomer design towards chemically recyclable polymers and served as a toolbox for fine-tuning the material properties via functionalization.

The experimental is poor. There appears to be many formatting errors, the ChemDraws are inconsistent and many are missing symbols, and the characterisation is poor for both small molecule products (^1H NMR spectra are not enough to show purity or prove composition – where are the other NMR experiments and either a HR-MS or elemental analysis?). When there is a ^{13}C spectrum, it is labelled as a “ ^{13}H NMR” such as in Figure S34! Did the authors not check this file after uploading? This poor quality suggests a lack of attention to detail from the authors.

Our response: We apologize for these mistakes and we have carefully proofread our new version of supporting information. For the known/reported small molecules, we showed their ^1H NMR spectra to confirm the structure and purity. For all the new compounds, we provided both ^1H NMR, ^{13}C NMR spectra and HR-MS data.

There also is a glaring lack of information on reaction kinetics and progress – conclusions are drawn from timepoints when known reactivity ratios and substitution patterns will suggest very different initiation and propagation rates, but no kinetics (or M_n vs conversion plots) are provided in either the main manuscript or the SI. The Van't Hoff plot is fine, but this isn't anything but a snapshot. The rest of the polymer characterisation is also lacking, with plots inconsistent in the DSC cycling (i.e. heat, cool, heat; lack of definition in peaks, spectra appear pasted in to generic scale diagram. Better, more consistent spectra, more clearly showing the desired features would be beneficial. The same could be said for the SEC, which again look copy/pasted in. Everything is repetitive and poorly presented, almost in an effort to make it longer rather than informative. This isn't a thesis, but a curated paper submitted to an exceptionally high impact journal.

Our response: We appreciate the reviewer's suggestion and have probed the polymerization kinetics of **M1–M8** by ^1H NMR spectroscopy. The conversion vs. time plots were summarized in Supplementary Figure 30. These monomers approached polymerization equilibrium within 40 mins at room temperature with a $[\text{M}]:[\text{Zn-1}]:[\text{I}]$ ratio of 1000:1:1. Similar polymerization activity were observed for **M1–M8**. Based on their similar kinetics, we would like to focus on their thermodynamic change caused by the substitution effect.

Supplementary Figure 30. The conversion vs. time plots of **M1–M8** with a $[\text{M}]:[\text{Zn-1}]:[\text{I}]$ ratio of 1000:1:1, $C = 1$ M and RT.

According to the preliminary kinetic results, these monomers exhibited fast polymerization rate even at a $[\text{M}]:[\text{cat.}]:[\text{I}]$ ratio of 1000:1:1. It's very difficult to monitor the polymerization process at a lower $[\text{M}]:[\text{cat.}]:[\text{I}]$ ratio (e.g. 200:1:1) due to the fast polymerization. Additionally, at the low catalyst loading of 0.1%, the polymerization performance was very sensitive to the impurity in the system. A trace amount of impurity could initiate undesired side reactions (chain transfer or initiation) or poison the catalyst, leading to *not* well-controlled polymerization feature.

We have revised the DSC figures with definition in peaks and presented them in a consistent format. We also removed some SEC traces as the reviewer suggested.

e.g.

Supplementary Figure 6 DSC curves for P(M15), $T_g = 61$, $T_m = 164\text{ }^\circ\text{C}$, $T_c = 122\text{ }^\circ\text{C}$.

In all, this is not an appropriate paper for Nature Communications, as while the volume of work is very high it is written almost exclusively for a polymer audience, it doesn't understand sustainability challenges in polymer recycling, it doesn't contextualise the results versus the literature, and instead simply presents masses of data in a repetitive way without showing new insights or impact. I would suggest the authors use the comments to revise the manuscript into a strong Macromolecules or Polymer Chemistry paper, as that is the appropriate audience for this work.

Our response: We sincerely welcome and appreciate these comments from the reviewer. We have tried our best to improve the quality of our manuscript based on the given comments.

Reviewer #3 (Remarks to the Author):

Reviewer's general comment: The manuscript applied substitution strategy to tune the thermodynamics of seven-membered ring caprolactone derivatives. The authors designed and prepared a series of alkyl substituted and spirocyclic acetal-functionalized caprolactones and examined their ROP behaviors by using zinc and yttrium based catalysts. They systematically studied the ROP thermodynamic of obtained cyclic monomers and demonstrated that the chemical recycling properties of resultant polymers depended on the size, position and structures of their respective substituents. Their finding demonstrated that such strategy can offer various opportunity to prepare circular close-loop polymer with high melting temperature and mechanical properties. This represented a systematic strategy to create a serial close-loop polyesters with tunable melting temperature and potential high performances polymers. These finding contributed important advances in the field of close-loop polymers, and it deserves publications in Nature Comm. Moreover, the manuscript is well-written and the conclusions are well supported by the experimental data. Thus, I recommend accept after some minor revisions outlined below:

Our response: Thank you very much for your kind comments and recommendation for publication in *Nat. Commun.*. We also thank your valuable comments that helped us improve the quality of the manuscript. We have revised the manuscript according to your specific comments listed below.

Reviewer's specific comments:

1. The symbol of Gibbs free energy should be italicized in Figure 1. I recommend that the values of T_c should be added in Figure 1a, that the effects of different fused-ring structure on T_c s can be compared more clearly.

Our response: Thank you for pointing out our oversight and we have added the values of T_c in Figure 1. The updated Fig.1 was copied below.

Fig. 1 Polymerization-depolymerization thermodynamic modification towards chemically recyclable polymers. Fused/bridged-ring strategy has been applied to increase the ceiling temperature (T_c) for PGDL system (relatively low T_c). In PCL system with relatively high T_c , substitution strategy was investigated.

2. In page 5 line 96, Table 1 and footnote of Table 2, the format of initial monomer concentration (like 1.0 M) was different from others (1M) in manuscript. In page 7 line 129, “83.0–137 kg mol⁻¹” also should be revised. In page 7 line 145, “M12” should be revised into “M15”. In Figure 4, the “Repolymerization” at first curve should be revised into “repolymerization”. In page 14 line 299, the type size should be revised. The abbreviation of reference 55 should be revised.

Our response: Corrections have been made. For number-average molecular weights determined by size exclusion chromatography (SEC), we would round the numbers to three significant figures.

3. Monomer M7 with geminal dimethyl substitutions on both β and δ positions cannot polymerize at RT. Did the author try different catalysts or ROP conditions such as low temperature? At least, decrease ROP temperature should be attempted.

Our response: We greatly appreciate the reviewer’s suggestion and have performed the ROP of M7 at low temperature. The results were summarized in Supplementary Table 2. M7 cannot polymerize under these conditions.

Supplementary Table 2 ROP results of **M7** at low temperature.^[a]

Run	[M]:[Zn-1]:[I]	Concentration (mol/L)	Time (h)	Conv. ^[b] (%)	T (°C)
1	100:1:1	1.0	28	0	0
2	100:1:1	2.0	24	0	0
3	100:1:1	2.0	24	0	-30
4	100:1:1	2.0	26	0	-40

[a] Condition: Catalyst = **Zn-1**, **M** = 100 mg, initiator (I) = *p*-tolylmethanol, solvent = THF.

Supplementary Figure 29 ¹H NMR spectra of the ROP mixture of **M7** (entries 1–4), *Solvent.

4. The M_n ,SEC of PM6 is obviously lower than theoretical value, while the distribution remains narrow (1.14). It may be ascribed to the multiple initiation caused by impurities rather than chain transfer. Corresponding MALDI-TOF MS of PM6 should be provided to further demonstrate the results.

Our response: Thank you for the helpful suggestion. We believe the reviewer is correct that the inconsistency of M_n ,SEC and M_n ,Calcd of P(**M6**) could be ascribed to the multiple initiation caused by impurities in the polymerization system. When we reperformed the ROP of **M6**, it's found that **M6** with an improved purity exhibited a similar polymerization reactivity with other monomers, approach 89% monomer conversion within 30 min. The resulting P(**M6**) showed M_n ,SEC of 208 and D of 1.05.

We have updated entry 7 in Table 2.

The microstructure of the resulting P(**M6**) was characterized by matrix-assisted laser desorption ionization time-of-flight (MALDI-TOF) mass spectrometry. The MALDI-TOF mass spectrum of the P(**M6**) sample produced by a [**M6**]:[**Zn-1**]:[**I**] ratio = 50:1:1 exhibited a clear one series of ion peaks, confirming the linear structures of P(**M6**) with **M6** as the repeat unit and *p*-tolylmethanol as the initiator.

Corresponding MALDI-TOF mass spectrum of P(**M6**) were copied below.

Supplementary Figure 7 MALDI-TOF MS spectrum of P(**M6**) produced by [**M6**]/[**Zn-1**]/[**I**] = 50/1/1 in 1 M at RT.

Supplementary Figure 8 Linear plot of m/z values (y) vs the number of **M6** repeat units (x).

5. According to the thermodynamic formula, the T_c of **M1** is calculated as 70 °C at 0.02M, which is lower than 140 °C. Hence, **PM1** should be completely degraded at 140 °C in 0.02M. However, the depolymerization of **PM1** only reached 69%. And Figure S128 showed obvious signals of side products that did not belong to recovered monomers and polymers. Hence, it may be a factor affecting the conversion. What are the structures of side products? Can the by-products be reduced by conducting the depolymerization at lower temperatures? Also, have the authors tried different degradation concentrations?

Our response: We appreciate the reviewer for this important remark. We totally agree with the reviewer that the formation of side products could affect the depolymerization conversions of **P(M1)**. We thought the side products were cyclic oligomers. When conducting the depolymerization at 70 °C, **P(M1)** was completely disappeared, but **M1** was recovered in 44% ($1.00/2.25 = 0.44$, as the following Supplementary Figure 126 shows) due to the formation of cyclic oligomers. We also screened the degradation concentration at 0.04 M and 0.01 M for the depolymerization of **P(M1)** (Supplementary Figure 127). Unfortunately, the formation of cyclic oligomers seems to be inevitable under these conditions.

We have provided explanation for Supplementary Figure 126. The formation of cyclic oligomers led to the low monomer recovery.

Supplementary Figure 9 Depolymerization study of P(M1) at 70 °C. ¹H NMR spectra of a) recycled M1 by the solution depolymerization (**Supplementary Table 1** entry 1, monomer conversion = $1.00 / (1.00 + 1.25) = 0.44$), top; b) starting M1 for comparison, middle; c) P(M1) ($M_n = 115$ kg/mol, $\mathcal{D} = 1.65$), bottom.

Supplementary Figure 10 Depolymerization study of P(M1) at different concentrations. ¹H NMR spectra of a) recycled M1 by the solution depolymerization (**Supplementary Table 2** entry 3, monomer conversion = $1.00 / (1.00 + 1.46) = 0.41$), top; b) recycled M1 by the solution depolymerization (**Supplementary Table 3** entry 2, monomer conversion = $1.00 / (1.00 + 0.16) = 0.86$); c) starting M1 for comparison, middle; d) P(M1) ($M_n = 81.9$ kg/mol, $\mathcal{D} = 1.68$), bottom.

6. The bp of toluene is about 110 °C, but the depolymerization reaction in toluene was performed at 140 °C. Please give more details in experiments of how to realize this.

Our response: We apologize for this confusing description. To clarify, 140 °C was the temperature of the bath and a pressure tube was used for these chemical recycling experiments.

We have provided these details in the supporting information.

“A pressure tube containing the purified polymer sample (20 mg) with 2 mol% Zn-1 in toluene (0.02 M) was sealed and heated to 140 °C (bath temperature) for 1 h under an argon atmosphere. After cooling back to room temperature, the reaction mixture was concentrated (evaporation in the watch glass or under vacuum) to give a colorless product, which was used for ¹H NMR analysis to determine the recycled monomer conversion.”

7. In page 8 line 164, I want to know whether the “[P(M)] = 20 mM in toluene” and “2 mol % Zn-1” was based on moles of polymers or structural units in polymers. In addition, the conversion rather than yield can be calculated according to in-situ ¹H NMR spectra.

Our response: We have provided more detailed information in this section and revised our statement related to yields to conversions as the reviewer suggest. “[P(M)] = 20 mM in toluene based on the moles of repeat units in polymers) with 2 mol % Zn-1 catalyst at 140 °C for 1 h”

8. The DSC curve of PM5 should be added to Figure b or c.

Our response: The DSC curve of P(M5) has been added into Fig 5b as the reviewer suggested.

Fig. 5 Thermal and mechanical properties of P(M)s. **a**, TGA curves of representative P(M)s. **b**, and **c**, DSC curves of representative P(M)s. **d**, and **e**, Strain-stress curves of representative P(M)s. Images showing the P(M13) film. **f**, Cyclic tensile testing of P(M10).

9. Did the polymers (PM8, PM10 and PM12) possess stiffness to undergo mechanical testing at 25 °C, because their T_g s are lower than 25 °C? It is interesting that these polymers with low T_g s have good mechanical properties, lack of crystallization behavior or crosslink structures.

Our response: The mechanical testing of P(M8), P(M10), and P(M12) was performed at room temperature (20–25 °C) which is slightly higher than their T_g values. We believe their excellent elastic behavior is attributed to the flexible structures and T_g values close to room temperature.

Reviewers' Comments:

Reviewer #1:

Remarks to the Author:

Overall, the authors have done a good job in addressing my comments, and I recommend its publication. I have a minor critique about their response on my comment 1, i.e., rationalization of such behavior from the molecular level rationalization. I appreciate the explanation on enthalpic using repulsion strain, but the entropic effect was explained with "increased order of the system", which is not an explanation but rather a restatement. In their next round revision, I encourage the authors to elaborate a little further on this aspect.

Reviewer #2:

Remarks to the Author:

While I appreciate the effort to improve the manuscript - the improved figures and additional kinetics are welcome - and an appropriate number of references have been added - I think the authors have missed the point of my desire for contextualisation.

Several of my paragraphs inspired no manuscript changes - and indeed the article remains as a conceptual panacea to "solve" the plastics crisis. The authors state that the lack of context is fine because "Ultimately, we have faith that a scalable, economically competitive, and chemically recyclable polymer system with properties comparable to today's plastic materials would be designed/discovered after massive efforts devoted to this research field."

Indeed this is my point. You do not have a scalable system - the catalysts aren't economically competitive, the monomer synthesis routes are expensive, no sustainability metrics are included, and the properties are not fully characterised to show they match today's plastics materials (i.e. weathering, permeability, processability). So the reader is left to compare this to other chemically recyclable monomer systems. A good example of this is the recent paper published by Chen - it deservedly is published in a high impact journal because it is disruptive, not iterative. This is below even many of the properties from the first generation system. It's a great Macromolecules paper, but this is just not an improvement on the state-of-the-art - these are not better chemically recyclable polymers than previously reported.

The team have also not contextualised that this proposed fate is realisable - they suggest that mechanical recycling is degradative. This is true, but it is exacerbated by polymers exactly like this. Contamination leads to down cycling.

I really wish the authors had actually properly addressed my comments on properly contextualising their work. The rest of the corrections seem fine.

Reviewer #3:

Remarks to the Author:

This manuscript has been greatly improved with carefully revision and all my comments have been addressed. The reviewer think this manuscript is acceptable for publication in Nature Communications.

Changes Made in the Revision and Responses to the Comments by the Reviewers

Reviewer #1 (Remarks to the Author):

Reviewer's comment: Overall, the authors have done a good job in addressing my comments, and I recommend its publication. I have a minor critique about their response on my comment 1, i.e., rationalization of such behavior from the molecular level rationalization. I appreciate the explanation on enthalpic using repulsion strain, but the entropic effect was explained with "increased order of the system", which is not an explanation but rather a restatement. In their next round revision, I encourage the authors to elaborate a little further on this aspect.

Our response: We would like to thank you for the positive reports and recommendation of the publication in *Nat. Commun.*

In this new version of manuscript, we have revised the related section as the reviewer suggested. "Substitution on CL could lead to a decrease in the flexibility of polymer chains, devoting significant loss in the conformational degrees of freedom for the resulting polymers."

Reviewer #2 (Remarks to the Author):

Reviewer's comment: While I appreciate the effort to improve the manuscript - the improved figures and additional kinetics are welcome - and an appropriate number of references have been added - I think the authors have missed the point of my desire for contextualisation.

I really wish the authors had actually properly addressed my comments on properly contextualising their work. The rest of the corrections seem fine.

Our response: We sincerely welcome and appreciate these comments from the reviewer. We revised the manuscript according to your comments listed below.

Several of my paragraphs inspired no manuscript changes - and indeed the article remains as a conceptual panacea to "solve" the plastics crisis. The authors state that the lack of context is fine because "Ultimately, we have faith that a scalable, economically competitive, and chemically recyclable polymer system with properties comparable to today's plastic materials would be designed/discovered after massive efforts devoted to this research field."

Our response: We apologize for our misleading statement and would like to clarify that we agree with the reviewer about concerns related to our synthetic system. However, we would like to explain the potential of this research field and manifest the significance of our work in this field. "a new chemically recyclable polymer with properties that are commensurate with commercial plastics could serve as an alternative and replace the traditional nondegradable plastics to address the accumulation of plastic waste." The main focus of this work is the structure–de/polymerizability and structure–property relationships, which could serve as a

guideline for the future monomer design towards chemically recyclable polymers and a toolbox for fine-tuning the material properties via functionalization.

Indeed this is my point. You do not have a scalable system - the catalysts aren't economically competitive, the monomer synthesis routes are expensive, no sustainability metrics are included, and the properties are not fully characterised to show they match today's plastics materials (i.e. weathering, permeability, processability). So the reader is left to compare this to other chemically recyclable monomer systems. A good example of this is the recent paper published by Chen - it deservedly is published in a high impact journal because it is disruptive, not iterative. This is below even many of the properties from the first generation system. It's a great *Macromolecules* paper, but this is just not an improvement on the state-of-the-art - these are not better chemically recyclable polymers than previously reported.

Our response: We appreciate the reviewer's comment.

For catalysts: we have performed additional ring-opening polymerization experiments using commercially available and cost-effective $\text{Sn}(\text{Oct})_2$ as catalyst. These monomers exhibited good polymerization activity with these catalysts, showcasing the high polymerizability of these CL-based monomers (Supplementary Table 1, entries 19–25).

For monomer synthesis: a single-step Baeyer-Villiger oxidation of cyclic ketones was exploited to construct a library of substituted caprolactones (**M1–M7**) on large scales (>10 g) with high yields (69–96%). Actually, the currently industrial synthesis of ϵ -caprolactone is via oxidation of cyclohexanone (*Industrial Organic Chemistry*, 4th ed., Wiley-VCH, Weinheim, 2003). Consequently, we believe this synthetic approach would be promising and feasible for large-scale monomer preparation.

For polymer characterization: we have thoroughly characterized the thermal and mechanical properties of the synthesized polymers and we do understand the material properties such as permeability and processability are important for practical applications. Unfortunately, as a polymer chemistry laboratory, we don't have enough experience or instruments to characterize these material properties at current stage, but we are excited to explore these material further and will certainly consider the reviewer's suggestion in the future study. This recent *Science* paper (doi/10.1126/science.adg4520) by Chen's group actually manifested the power of the *gem*-disubstituent strategy in the four-membered lactone system. In this work, a spirocyclic substitution strategy was applied to expand the library of geminal disubstituted caprolactone-based monomers. Notably, this new class of monomers (**M9–M17**) inherited the efficient polymerizability and excellent chemical recyclability from geminal dimethyl-substituted monomer (**M5**). More impressively, the spirocyclic substitution strategy imparted the resulting polymers with tunable properties by the observation of thermal transformation from amorphous to semicrystalline and mechanical transformation from brittleness to ductility, which will be vital for optimizing their performance in future applications from elastomers to plastics.

Overall, we would like to emphasize that this fundamental work is aimed to provide detailed and systematic understanding of the factors that affect thermodynamics and material properties and serve as a guideline for future monomer design in this field.

The team have also not contextualised that this proposed fate is realisable - they suggest that mechanical recycling is degradative. This is true, but it is exacerbated by polymers exactly like this. Contamination leads to down cycling.

Our response: We understand the reviewer's concern and have realized this chemical recycling to monomer (CRM) strategy might still face many issues such as plastics separation and contamination. However, recent works have demonstrated the ability of the CRM strategy to isolate pure monomer from complex mixtures and avoid the need for time- and cost-intensive separation processes (e.g. *Science* **373**, 783-789, (2021) and *Sci. China Chem.* **66**, 251-258, (2023)). Nevertheless, we believe this CRM strategy provided an attractive approach to address the plastic crisis and our fundamental work related to the structure–de/polymerizability and structure–property relationships serve as a guideline for the future monomer design towards chemically recyclable polymers.

Reviewer #3 (Remarks to the Author):

Reviewer's comment: This manuscript has been greatly improved with carefully revision and all my comments have been addressed. The reviewer think this manuscript is acceptable for publication in *Nature Communications*.

Our response: Thank you very much for your kind comments and recommendation for publication in *Nat. Commun.*.

Reviewer #2 (Remarks to the Author):

The authors remain confused as to what my comments are asking them to address, and thus I remain frustrated by the presentation of their work. I reiterate that this would be a lovely *Macromolecules* paper, but it is not something I would expect to see in *Nature Communications*.

The authors suggest that they are unable or unwilling to address key polymer properties (because they are polymer chemists, in their words) and unable or unwilling to measure sustainability metrics (they suggest this is because their route is the same as poly(ϵ -caprolactone)). If they had bothered to look up sustainability metrics for PCL they would realise it has an exceptionally high carbon footprint due to the same production method they are using - and of course this is from a relatively low carbon hexanone. The much higher footprint of their substituted monomers would preclude any commercial application without severe environmental damage. The authors also suggest that there is no damage to mechanical recycling because CRM works on mixed feedstocks. This again misses the point - contamination of mechanical recycling is the concern, not CRM.

The authors have not properly addressed my comments. As the third time reading the manuscript it remains overzealous in its presentation of the results. The structure-property relationships are fine, but the 'best' of these polymers remain below best-in-class CRM systems. Without a broad, authentic presentation I cannot recommend this for publication.

Changes Made in the Revision and Responses to the Comments by the Reviewers

Reviewer #2 (Remarks to the Author):

Reviewer's comment: The authors remain confused as to what my comments are asking them to address, and thus I remain frustrated by the presentation of their work. I reiterate that this would be a lovely Macromolecules paper, but it is not something I would expect to see in Nature Communications.

The authors have not properly addressed my comments. As the third time reading the manuscript it remains overzealous in its presentation of the results. The structure-property relationships are fine, but the 'best' of these polymers remain below best-in-class CRM systems. Without a broad, authentic presentation I cannot recommend this for publication.

Our response: Thank you for taking your time to give us your valuable comments. We believe our work can shed light on the monomer design strategy towards chemically recyclable polymers and provide concrete guideline for the development of future chemically recyclable materials. As the revision of our initial manuscript was recognized by other reviewers, we would be grateful if the reviewer could reconsider our manuscript.

The authors suggest that they are unable or unwilling to address key polymer properties (because they are polymer chemists, in their words) and unable or unwilling to measure sustainability metrics (they suggest this is because their route is the same as poly(ϵ -caprolactone)). If they had bothered to look up sustainability metrics for PCL they would realise it has an exceptionally high carbon footprint due to the same production method they are using - and of course this is from a relatively low carbon hexanone. The much higher footprint of their substituted monomers would preclude any commercial application without severe environmental damage. The authors also suggest that there is no damage to mechanical recycling because CRM works on mixed feedstocks. This again misses the point - contamination of mechanical recycling is the concern, not CRM.

Our response: We apologize for any inappropriate statement caused by our previous response. To clarify, we understand the reviewer's concern of our monomer synthetic method regarding to sustainability metrics, but we would like to point out that an ideal CRM system could provide a circular economy and serve as a new class of sustainable polymer materials. This work actually evaluated substitution effects and structure-property relationships in the ϵ -caprolactone system to guide the design of monomers for the preparation of chemically recyclable polymers. Therefore, this study focused on the monomer design principle and structure-property relationships. In our future study, we will characterize the key material properties including permeability and processability besides thermal and mechanical properties. Lastly, we fully agree with the reviewer that contamination of mechanical recycling could be the concern but would like to re-emphasize the focus and significance of our work. Our proof-of-concept laboratory study is aimed to provide

a solid support in principle for monomer design strategy and serve as a guideline to drive the development of chemically recyclable polymers.